# FOUNDATION MODELS FOR BOOLEAN LOGIC

## ABSTRACT

Boolean logic is fundamental to solving various computational problems, such as Boolean satisfiability (SAT) and model counting, but existing machine learning (ML) approaches for automating algorithm design are computationally expensive and data-intensive. We propose the first foundation model for Boolean logic, leveraging a multi-task dataset of one million instances spanning sixteen tasks and using graph neural networks (GNNs). We evaluated the generalization of the foundation models on held-out tasks; we found that models fine-tuned from the foundation model were substantially more sample efficient and converged much faster than models trained from scratch. We identified a number of crucial design components for training these models, in particular the choice of normalization layer. We showed that a hybrid of different normalization techniques across layers is much more effective than any single normalization layer.

## 1 INTRODUCTION

Boolean logic—binary operations $\wedge$ (AND), $\vee$ (OR), and $\neg$ (NOT) over Boolean variables—is a fundamental mathematical language for describing many real-world settings. There are a variety of computational problems over logical formulae commonly solved in practice:

- Boolean Satisfiability (SAT): Determining if there exists a satisfying assignment. Example: Finding valid channel assignments for TV stations in spectrum auctions (Fréchette et al., 2016).
- Model Counting (#SAT): Calculating the number of satisfying assignments. Example: Determining the probability that a sequence of actions achieves a goal in probabilistic planning (Domshlak & Hoffmann, 2006).
- Unsatisfiable Core Extraction (unsat core): Identifying the smallest set of variables that prove no satisfying assignment exists. Example: Finding the minimal conflict in circuit configurations to aid system debugging (Sülflow et al., 2008).

Despite their computational hardness, efficient heuristic algorithms have been developed over decades of empirically driven research to solve these problems at scale. The performance of different algorithms often depends on the specific structure of problem instances, leading to designs tailored for "typical-case" scenarios. This creates a data-dependent algorithm design challenge: how can we design efficient algorithms for particular distributions of problems? Manual approaches are limited and time intensive, naturally steering us toward machine learning to leverage the rich structure of formulae. Just as machine learning has surpassed human capabilities in tasks like image recognition by identifying complex patterns, it can similarly be employed to discover effective algorithms for Boolean logic.

Traditionally, leveraging machine learning for algorithm design has relied on computing hand-crafted instance features based on expert domain knowledge. For example, practitioners build *algorithm selector* models that use such features to make a per-instance choie among a portfolio of off-the-shelf algorithms, leveraging their complementary strengths. Features used in Boolean logic range from simple metrics like problem size to complex ones like the diameter of the variable-clause graph or statistics from short probing runs of local-search and CDCL solvers (Xu et al., 2008). While powerful, such features can be expensive to compute and difficult to transfer to new domains.

Recent work has demonstrated the promise of leveraging modern "end-to-end" ML techniques to learn features directly from data. To give some examples of approaches that achieved state-of-the-

art performance in given settings, Selsam & Bjørner (2019) predicted unsatisfiable cores to guide branching decisions, Wang et al. (2021) predicted the polarity of backbone variables to choose variable assignments in tree search, and Cameron et al. (2024) learned branching policies via reinforcement learning to minimize downstream decisions. A key idea unifying all of these approaches is a reliance on message-passing architectures, such as graph neural networks (GNNs) (Scarselli et al., 2008) and exchangeable nets (Hartford et al., 2018). Such architectures impose a helpful inductive bias, corresponding to the invariances of Boolean logic in conjunctive normal form (CNF): (1) logical equivalence under reordering of clauses and literals, and (2) variability in the number of literals and clauses.

The downside of these approaches is that they are extremely data hungry. For example, all of the approaches described above required many CPU years of computation to generate training data. To achieve strong performance, practitioners must gather huge datasets for specific prediction tasks, where the offline computation costs can be prohibitive. In other fields, large pretrained models trained on massive, multi-task datasets—known as foundation models—can be fine-tuned to specific applications to massively decrease training costs. (Betker et al., 2023; Achiam et al., 2023). These models leverage shared information across multiple tasks to learn richer and more generalizable representations.

For the first time, we developed foundation models for Boolean logic, demonstrating strong fine-tuning performance on held-out tasks. We compiled a massive dataset of one million small instances encompassing ten different categories (sixteen tasks in total) of Boolean logic-based tasks: four well-known computational problems (satisfiability, backbone, unsat core, and model counting), a linear programming relaxation, a branch prediction task based on reinforcement learning, three that are statistics of probing runs from SAT solvers (DPLL, local search, and survey propagation), and one predicting graph structure. Notably, some tasks are only applicable to either satisfiable or unsatisfiable instances—backbones are defined solely for satisfiable instances, while unsatisfiable cores apply exclusively to unsatisfiable ones. We trained ten different Graph Neural Network (GNN) foundation models, each with a distinct held-out task. Foundation models were consistently more data efficient and converged faster when fine-tuned on the same held-out tasks relative to models trained from scratch.

The major challenge of building a foundation model is to find one architecture that works well across many diverse kinds of tasks. We found that architectures that performed well on one task can perform poorly on another. A major contribution of this work was finding an architecture that performed well across all tasks. First, we found that all commonly used normalization layers (i.e, layer norm, batch norm, graph norm) had some failure cases. Layer norm was unable to learn graph-level tasks at all and for batch and graph norm we observed erratic training behaviour, which we attributed to the high-variance batch statistics. We found that using a hybrid norm—batch norm for GNN layers and layer norm for the feed-forward model— substantially improved performance and training efficiency. We provided some empirical evidence that the success of hybrid norm is in its ability to avoid both the numerical instability that can occur with batch norm as well as the over-smoothing of node embeddings we observed with layer norm. The success of this hybrid norm approach could be of significant interest to the GNN community more broadly. We also found that both dropout and mean pooling often substantially degraded performance and we observed consistent performance improvement by adding self-attention over node embeddings between GNN layers. We found using sum pooling and turning off dropout was the best configuration for all tasks.

We used a hydrid GNN transformer models based on GPS++ (Masters et al., 2022), which is among the state-of-the-art models for standard graph benchmarks. Our final foundation model had eight layers and 122 million parameters, with each layer comprising a GPS message-passing component followed by self-attention over node embeddings. We add a feed-forward head for each of the sixteen total tasks: seven graph-level (one classification, six regression) and nine node-level (three classification, eight regression).

This work serves as a major step towards building foundation models for Boolean logic. We found an architecture that works well across a wide-variety of tasks which will make future work much more accessible. We envision a future with large, GPT-like pretrained Boolean logic models with billions of parameters that can be fine-tuned for a wide range of tasks.

## 2 PRELIMINARIES AND RELATED WORK

### 2.1 HAND-CRAFTED ML

Prior to Selsam et al. (2019)'s first attempt to learn a model to represent Boolean logic, hand-crafted features were exclusively used for making per-instance predictions (typically predicting solver running times) and still dominate the research today. For example, algorithm selection is still an active area of research and it was only until very recently (Zhang et al., 2024; Leeson & Dwyer, 2024) that GNNs have been applied to that problem. Nudelman et al. (2004) introduced a set of hand-crafted features that were expanded by Xu et al. (2008), later again by Hutter et al. (2014), and have been recently upgraded to be more informative (e.g., smartly choosing timeouts on probing runs) (Shavit & Hoos, 2024). They have been proven to be effective for building empirical hardness models of algorithms (Hutter et al., 2014) and for algorithm selection (Xu et al., 2008; Lindauer et al., 2015). These features were derived from various sources: known heuristics (e.g., the ratio of positive to negative clause occurrences and per-variable statistics), tractable subclasses (such as the proximity to Horn formulae), graph-based features (like properties extracted from the clause-variable incidence graph), and other proxies for problem complexity (including statistics about the progress of SAT solvers and linear programming relaxations of the SAT problem). The computational complexity of these features spans a wide range, from trivial calculations (like determining the size of the problem) to more computationally intensive tasks (such as computing LP relaxations or extracting specific graph-based features, which can be roughly cubic in complexity). Many other combinatorial problems like MIP and TSP (Hutter et al., 2014) have relied on similar hand-crafted features.

### 2.2 GNNS

The ML for Boolean logic community has converged on representing logical formulae as graphs and using GNNs. Selsam et al. (2019) pioneered this by encoding CNF SAT formulae as graphs with variables, clauses, and true/false literals as nodes, connecting variables and clauses if a variable participates in a clause, and linking literals to their variables. Their approach used two message-passing operations—between clauses and variables, and between variables and literals—achieving high accuracy in predicting satisfiability and deriving solutions for small random SAT problems. Cameron et al. (2020) instead represent CNF as an *exchangeable* matrix, using an architecture equivalent to a bipartite variable-clause graph (Hartford et al., 2018). For broader insights on representing combinatorial problems as graphs, see Boisvert et al. (2024).

GNNs have been used to predict unsatisfiable cores (Selsam & Bjørner, 2019), predict satisfiability (Cameron et al., 2020), predict branching variables (Kurin et al., 2019; Cameron et al., 2024), and for algorithm selection (Zhang et al., 2024; Leeson & Dwyer, 2024). Perhaps the closest work to our own is on predicting backbones, where Wang et al. (2021) first pretrained on a large dataset of backbones from small instances and fine-tuned on larger instances. Also relevant is the recent work of Li et al. (2023) who compiled a large benchmark of instances and tasks to benchmark GNN performance in Boolean logic.

### 2.3 MULTI-TASK LEARNING AND FOUNDATION MODELS

The advent of foundation models—large-scale pretrained models capable of being fine-tuned for a multitude of downstream tasks—has revolutionized fields like natural language processing (Brown, 2020) and computer vision (Betker et al., 2023). Typically, the multi-task aspect of foundation models in vision and language is implicit. For example, Large Language Models (LLMs) like GPT are primarily framed to predict the next word in a sequence; this objective inherently requires solving a variety of implicit tasks depending on the context provided within the input text. Multi-task learning (MTL) explicitly defines and optimizes multiple tasks simultaneously for a given input (Yu et al., 2024). In contrast to foundation models where tasks are inferred from context, MTL frameworks require distinct task definitions integrated into the model architecture. Typically, this involves designing models with shared layers that learn a common representation, alongside task-specific heads that handle individual objectives. As one notable example, Beaini et al. (2023) built a multitask foundation model over a variety of molecule prediction tasks and datasets. Our implementation is based off their `graphium` package.

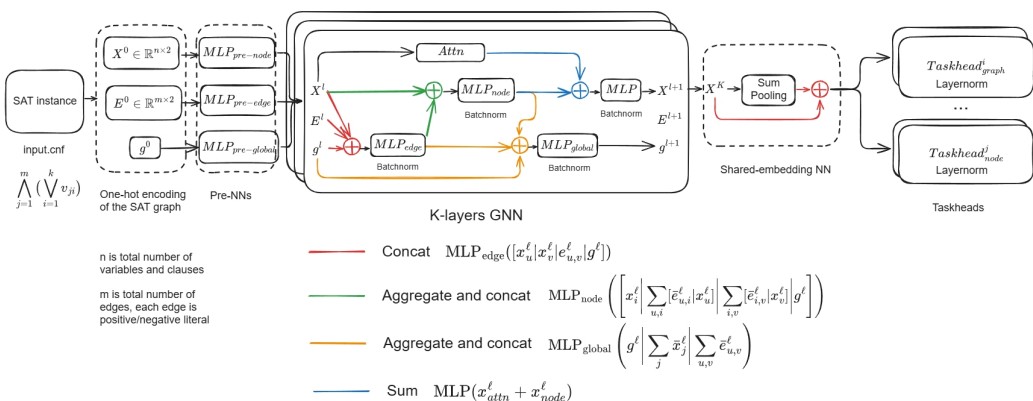

Figure 1: Our foundation model architecture. One-hot encoded SAT instance goes into feed-forward pre-NN encoders followed by a sequence of hybrid MPNN+transformer layers. GNN output then goes through a global pooling and concat operation across nodes, and feed into feed-forward task heads for different tasks.

## 3 METHODS: END-TO-END BOOLEAN LOGIC MODELLING

Given a set of $k$ tasks $T = \{T_1, ..., T_k\}$ and instance distribution $\mathbb{P}$, our goal is to learn a function $\phi^*$ that minimizes the mean over $|T|$ loss functions

$$\mathcal{L}(\phi; \mathbb{P}) = \frac{1}{|T|} \sum_{i=1}^{|T|} \mathcal{L}_i(T_i, \phi).$$

We define losses for each task based on its category: mean squared loss for regression tasks and cross-entropy loss for classification tasks, averaged over nodes if it is predicting per node metrics. For detailed description of the losses, see Appendix B.

Instances in our case are Boolean logical formulae in conjunctive normal form (CNF). We represent CNFs with the well-known and lossless clause-variable bipartite graph, allowing us to model $\phi$ as a GNN which takes a graphical representation directly as input. We one-hot encode the intut CNF and pass the encoded node and edge embedding matrices $\mathbf{X}^0$ and $\mathbf{E}^0$ as inputs to the model. For a detailed description of how we encoded the CNF, see Appendix C.

GNNs involve a sequence of message-passing operations over nodes/edges, where a given node/edge representation is updated by aggregating (i.e., any commutative function) over its neighbouring nodes/edge representations and sending the aggregation through a multi-layer perceptron (MLP). Our GNN instantiation is adapted from GPS++ (Masters et al., 2022). Our network takes as input a graph $(\mathbf{X}^0, \mathbf{E}^0, g^0)$, where $g$ is a graph embedding that is represented as a "virtual" node connected to every other node. Our network consists of (1) a pre-GNN node and edge encoder that learns an embedding for each node/edge type followed by (2) a sequence of message-passing layers that iteratively update node and edge embeddings, (3) a shared-embedding layer that pools and concats graph and node level embeddings and (4) $|T|$ task heads that map the node embedding representation down to the target shape of the task. See figure 1 for a visualization of our end-to-end model architecture and Appendix D for a detailed description of each layer.

## 4 EXPERIMENT SETUP

### 4.1 DATASET

We built a dataset of one million uniform-random 3SAT instances at the solubility phase transition (Cheeseman et al., 1991), each with 100 variables. This allowed us to generate an arbitrary number of challenging training examples. The computing cost of gathering target labels can scale exponen-

tially with instance size, and model training resources scale linearly in both memory and time with the size of the formulae. By training our foundation model on smaller formulae, we were able to train much larger models and we were able to collect much more training data.

We compiled sixteen prediction tasks spanning ten different categories, all of which have been previously studied in the context of machine learning for Boolean logic. Four of these tasks—unsat core, backbone, RL branching, and model counting—have been used to directly improve SAT solver performance by informing branching decisions during tree search. One task, satisfiability, has been extensively studied as a testbed for end-to-end learning on Boolean formulae. The remaining five tasks have been employed as computationally inexpensive features for meta-algorithmic approaches to solving SAT, such as algorithm selection. Below, we describe for each task (1) its prior use in machine learning for Boolean logic and (2) how we computed its ground truth labels. For formal definitions of each task, please refer to Appendix A.

**Predicting satisfiability** Selsam et al. (2019) were the first to demonstrate how GNNs could be used for end-to-end learning for Boolean logic by predicting satisfiability. Cameron et al. (2020) later showed how GNNs could beat expert hand-engineered features for SAT prediction and there have been several later follow up works (Chang et al., 2022; Li et al., 2023). We used the model counting computation below to identify satisfiability. If the model count was zero, the instance is usatisfiable, otherwise it is satisfiable.

**Model counting** Vaezipoor et al. (2021) train a neural network to make branching decisions to solve model counting. We used the Sharpsat (Thurley, 2006) solver to compute model counts.

**Backbone** Wang et al. (2022) predicted backbones which they used to assign polarity to branching variables in CDCL solvers. We used the Cadiback (Biere et al., 2023) solver to compute backbones. Only defined for satisfiable instances.

**Unsatisfiable Core** Selsam & Bjørner (2019) learned a GNN model to predict unsatisfiable cores which they then used to make branching decisions in CDCL solvers (i.e., branch of the variable predicted to be most likely to belong to smallest core); they acheived state-of-the-art performance. We used the z3 program (De Moura & Bjørner, 2008) to compute approximately minimal unsatisfiable cores. Only defined for unsatisfiable instances.

**RL-based branching** Cameron et al. (2024) used an RL procedure to learn a model to estimate the relative effectiveness of branching on each variable (which they then used as a branching policy to improve SAT solvers). We ran MCFS at the root of the tree for 100,000 lookaheads with identical settings to Cameron et al. (2024) and measured variable counts and tree-size estimates. Only defined for unsatisfiable instances.

**Instance-level properties** Leyton-Brown et al. (2003); Xu et al. (2008); Hutter et al. (2014) developed a number of features for representing Boolean logic that have been used to build prediction models to predict solver running times (Hutter et al., 2014), select amongst a set of algorithms Xu et al. (2012), and to configure SAT solvers (Hutter et al., 2011). We partitioned these features into five tasks: graphical structure, linear programming relaxation, and statistics from probing runs of three types of SAT solvers (local search, dpll probing, survey propagation). We use the feature generation script from (Hutter et al., 2014).

**Variable/Clause properties** The features from (Hutter et al., 2014) are on the intance level and many are aggregations across variables or clause statistics. We developed finer-grained variations of these features at the level of variables and clauses as well as other features that don't make sense at an instance level (e.g., number of times variable is flipped in local search).

We computed the ground truth for each task on every instance, except for tasks that are defined only for satisfiable/unsatisfiable instances. We used 2.40 GHz 2 x AMD Rome 7532 CPUs with 8GB of RAM. The dataset required 20 CPU years in total to label. We divided our dataset into an 80:10:10 train, validation, and test split.

We will make this dataset publicly available on hugging face to help facilitate further research. We believe that this dataset can serve as an excellent test bed for evaluating various GNN approaches in the context of Boolean logic.

## 4.2 Within Distribution Task Generalization

We evaluated our foundation model first on how effectively we could fine-tune to new tasks. We evaluated how (1) data-efficient and (2) training-time efficient fine-tuning was relative to training from scratch. We then evaluated whether efficiency gains from fine-tuning were a consequence of the diversity of pretraining tasks. We compared fine-tuning from foundation model vs fine-tuning from pretrainged models from single tasks.

**Data Efficiency** We trained ten foundation models with each task category held out on the all one million instances. We then randomly subsampled the training set at 100, 1000, 10000, and 100000 instances. For each dataset size and task category, we trained two models with identical architecture: (1) fine-tuning from corresponding foundation model for held-out task and (2) training from scratch.

**Faster Convergence** For each task category, we evaluated fine-tuning against training from scratch on the full dataset. We ran two variants of finetuning in each case. One where we fine-tuned all of the parameters, and another where we froze the shared architecture and only trained the task head. This latter setting is much less demanding on GPU resources, especially for much larger Boolean logical formulae we might encounter in practice. We measured validation performance at regular intervals and compared training-time efficiency.

**Finetuning from single-task pretraining** For five different tasks (SAT, #SAT, bakcbone, unsat core and RL branching), we built pretrained single-task models. We fine-tuned each task from each pretrained model and compared performance relative to fine-tuning from the foundation model. We froze the shared architecture in each fine-tuning experiment.

## 4.3 Out-Of-Distribution Generalization

We fine-tuned on evaluation to seven new distributions, three of which are non random. The non-random distributions are small-world graph colouring Hutter et al. (2014), quasi group completion Hutter et al. (2014), and spectrum repacking Fréchette et al. (2016). The other are uniform random 4SAT, uniform random 5SAT, and controlled and minimal backbone which are random instance with controlled backbone. We also fine-tune on eight larger size-datasets from the same distribution on variable 150-600 at intervals of 50 variables.

## 4.4 Model Training

We pretrained the model using a two-layer MLP for edge and node encoding, followed by eight message-passing layers and a two-layer MLP for each dataset-specific head. Each dataset was assigned a unique MLP head appended to the shared message-passing layers. The network used leaky ReLU activations and maintained 64-dimensional embeddings for nodes and edges. Optimization employed the Adam optimizer (Kingma & Ba, 2014) with a learning rate of $0.0001$, batch normalization (used with batching during validation to reduce high variance) in GNN layers, and layer normalization in MLP layers. Training used a batch size of 20 (80,000 nodes + edges), with batches sampled uniformly at random. Losses were masked for undefined or missing tasks (e.g., backbone loss on unsat instances). Pretraining, fine-tuning, and frozen fine-tuning runs were allocated 24, 6, and 2 hours, respectively. Performance was evaluated on a validation set per epoch, with the best model tested on a held-out test set. Experiments ran on A100 GPUs.

## 5 Results

### 5.1 Within distribution task generalization

Table 1 shows the improvement in fine-tuning on each hold-out task on the pretraining instance distribution on a small subset of the training set (1000 examples). Except when performance did not exceed the trivial baseline for both approaches (DPLL Probing, Var LP, Local Search), fine-tuning performed at least as well as training from scratch. This is most prominent for the four tasks that correspond to NP-hard computational problems. Fine-tuning had 10%, 8%, and 1% better accuracy in predicting satisfiability, backbone variables, and unsat core, respectively. It also performed three times better in terms of $r^2$ value for predicting model counting. Performance tended to be poor

on tasks from probing runs of SAT solvers (DPLL, Local Search). We suspect that these tasks are fundamentally difficult to predict. For a homogeneous distribution like uniform random, differences in these probing statistics across instances are likely to be dominated by noise. We also note the improvement in the RL-for-branching task (4% better accuracy in predicting a variable in the top 10). The loss differential appears minimal but in fact is a substantial difference. The target is the distribution of Q values across the actions (variables) in an RL procedure, which tends to be very close to uniform because it takes a lot of samples to pull apart the actions. Small differences in Q-value predictions are meaningful but not captured well by the magnitude of the cross entropy loss.

| Held-out Task | Type | Metric | Fine-tuning | | From Scratch | |
|---|---|---|---|---|---|---|
| | | | Loss | Metric | Loss | Metric |
| SAT | Graph Classification | Accuracy | **0.538** | **0.738** | 0.632 | 0.644 |
| Backbone | Node Classification | Accuracy | **0.832** | **0.619** | 0.953 | 0.535 |
| Unsat Core | | | **0.134** | **0.902** | 0.14 | 0.895 |
| RL Branching | Node Selection | Top-10 Accuracy | **4.6084** | **0.396** | 4.6086 | 0.359 |
| Var Structural | | | 0.990 | **0.099** | **0.989** | 0.099 |
| Clause Structural | | | **0.431** | **0.972** | 0.434 | 0.971 |
| Var Local Search | Node Regression | R2 Score | **1.771** | **0.057** | 1.800 | 0.041 |
| Clause Local Search | | | **0.494** | **0.705** | 0.612 | 0.634 |
| Var LP | | | **0.146** | -0.001 | 0.146 | **0.004** |
| Clause LP | | | **0.088** | **0.796** | 0.171 | 0.602 |
| #SAT | | | **1.168** | **0.327** | 1.551 | 0.107 |
| Local Search | | | **2.878** | **-16.749** | 11.155 | -67.792 |
| LP | Graph Regression | R2 Score | **1.086** | **0.044** | 1.128 | 0.007 |
| DPLL Probing | | | 0.993 | -0.134 | **0.886** | **-0.011** |
| Survey Propagation | | | **0.369** | **0.274** | 0.424 | 0.166 |
| Graph Structural | | | **0.529** | **0.147** | 0.622 | -0.003 |

Table 1: Performance comparison between fine-tuning and training from scratch on each task held out for a subsampled training set of 1000 examples.

It is very common in practice to have just a few hundred or thousand instances to train from (e.g. Bischl et al. (2016)) so performance at these dataset sizes is much more indicative of real-world performance. Figure 2a shows the complete results of the data efficiency experiments at the four orders of magnitude of dataset size. The base of the y-axis represents performance of the trivial baseline (random guessing for classification and predicting the mean for regression). In many cases, fine-tuning was an order of magnitude more data efficient, achieving better performance at 1000 examples than training from scratch achieved with 10,0000 examples. Satisfiability prediction was the most striking; fine-tuning from 100 examples outperformed training from scratch on 100,000 examples. In almost all cases, training from scratch eventually reached fine-tuning performance with a sufficiently large training set; datasets of this size are unlikely to be readily available for particular downstream applications.

We now show that fine-tuning performance also tended to converge much faster. Figure 2b shows validation performance over training runs comparing fine-tuning and training from scratch on the full dataset. For all four of the NP-hard computational tasks, fine-tuning is more than an order of magnitude faster; performance of fine-tuning after 1000 steps exceeded performance of training from scratch after 10,000 steps. The plot also shows performance of fine-tuning with the shared parameters frozen and just the task-head parameters are learnable. In many cases, this frozen model exceeds the performance of training all parameters from scratch before 10,000 steps.

Next, we evaluated whether fine-tuning from single-task pretrained models could achieve similar results to pretraining on all tasks. We found that in general that is not the case suggesting that the combination of pretraining tasks is leading to a more generalizable representation. See Table 2 for results. The left-most column lists the pretrained models and each column represents the tasks we are fine-tuning to. In every case, the foundation model trained on all but the fine-tuning task achieved best performance.

(a) Performance as a function of training set size.     (b) Validation performance over training run.

Figure 2: Performance of fine-tuning and training from scratch on each held-out task.

We also observed a clear distinction between tasks trained only on unsatisfiable instances (unsat core, RL branching) and the others. SAT, model counting, and backbone were mutually complementary but models trained on these tasks tended to fine-tune poorly to tasks on unsatisfiable tasks and vice versa. This is evidence that predictive features are very different between satisfiable and unsatisfiable instances.

| | **Held-out Tasks** | | | | |
|---|---|---|---|---|---|
| **Model** | **SAT** Acc | **# SAT** $R^2$ | **Backbone** Acc | **Unsat Core** Acc | **RL Branching** Top-10 Acc |
| Foundation | 0.743 | 0.432 | 0.614 | 0.894 | 0.408 |
| Sat | | 0.362 | 0.586 | 0.889 | 0.388 |
| Model Counting | 0.727 | | 0.592 | 0.872 | 0.385 |
| Backbone | 0.721 | 0.346 | | 0.886 | 0.363 |
| Unsat Core | 0.685 | 0.208 | 0.552 | | 0.406 |
| RL Branching | 0.630 | 0.091 | 0.522 | 0.890 | |

Table 2: Performance for finetuning task heads with frozen graph layers on six models (one foundation model trained on all but one task and five models trained on single tasks) and five held-out tasks. Colors are normalized by the max (green) and min (red) metric of each task. Foundation model outperformed all single task models.

## 5.2 OUT-OF-DISTRIBUTION GENERALIZATION

Table 3 shows out-of-distribution fine-tuning performance for 15 different settings. Finetuning consistently showed better performance. In the three nonrandom distributions, fine-tuning achieved 0.3 (vs. 0.11) $r^2$ for model counting (QGC), 0.992 (vs 0.986) auroc for unsat core (SATFC), and 0.09 (vs -0.07) $r^2$ for model counting (SWGC) . We also showed a consistent fine-tuning improvement for upward-size scaling for SAT prediction. For example, for 550 variables, fine-tuning achieved 85% accuracy compared 71% from training from scratch.

## 5.3 NORMALIZATION

We found that the performance of the foundation model was sensitive to the choice of specific combinations of normalization layers. We evaluated the performance of the foundation model for different normalization techniques: batch normalization (Ioffe & Szegedy, 2015) , layer normalization (Ba et al., 2016) and what we call *hybrid normalization*, which uses batch normalization in GNN and layer normalization in feed-forward head networks. We show in Table 4 that hybrid normalization outperformed all other normalization techniques for most Boolean logic tasks when training from scratch.

| Distribution | Mean #Var | Task | Metric Type | Foundation | | From Scratch | |
|---|---|---|---|---|---|---|---|
| | | | | Loss | Metric | Loss | Metric |
| Uniform Random 3SAT | 150 | SAT | Accuracy | **0.492** | **0.759** | 0.600 | 0.660 |
| | 200 | | | **0.436** | **0.793** | 0.594 | 0.662 |
| | 250 | | | **0.451** | **0.788** | 0.591 | 0.682 |
| | 300 | | | **0.439** | **0.802** | 0.577 | 0.694 |
| | 350 | | | **0.416** | **0.823** | 0.566 | 0.698 |
| | 400 | | | **0.448** | **0.784** | 0.582 | 0.666 |
| | 550 | | | **0.411** | **0.854** | 0.572 | 0.711 |
| | 600 | | | **0.394** | **0.829** | 0.555 | 0.712 |
| Uniform Random 4SAT | 90 | Backbone #SAT | Accuracy R2 score | **0.676** **0.508** | **0.727** **0.407** | 0.731 0.635 | 0.713 0.258 |
| Uniform Random 5SAT | 65 | #SAT | R2 score | 0.201 | 0.368 | **0.191** | **0.399** |
| Backbone Minimal Subinstance | 100 | Backbone #SAT | Accuracy R2 score | **1.001** **2.698** | 0.504 **0.209** | 1.006 3.452 | **0.509** -0.011 |
| Controlled Backbone Size | 100 | Backbone #SAT | Accuracy R2 score | **0.798** **0.967** | **0.633** **0.329** | 0.856 1.304 | 0.596 0.095 |
| Quasi Group Completion (QGC) | 1299 | #SAT | R2 score | **1.188** | **0.315** | 1.539 | 0.112 |
| Spectrum Repacking Problem (SATFC) | 782 | Unsat Core | AUROC | **0.116** | **0.992** | 0.128 | 0.986 |
| Small-world Graph Colouring (SWGC) | 1332 | #SAT | R2 score | **1.884** | **0.093** | 2.231 | -0.074 |

Table 3: Out-of-distribution finetuning on seven new distributions and eight upward-size generalization datasets of the same pretraining distribution.

| Normalization | SAT Acc | # SAT $R^2$ | Backbone Acc | Unsat Core Acc | RL Branching Loss[*] |
|---|---|---|---|---|---|
| Batch | 0.646 | 0.130 | 0.535 | 0.867 | 0.0460878 |
| Layer | 0.651 | 0.137 | 0.537 | 0.877 | 0.0460876 |
| Hybrid | **0.756** | **0.448** | **0.634** | **0.904** | **0.0460860** |

Table 4: Foundation model performance (val loss) for each normalization layer type: batch normalization, layer normalization, and hybrid normalization (batch norm for GNN layers + layer norm for feed-forward heads). [*] Had issue with RL branching metric; We will have it for camera ready.

Batch normalization and layer normalization are two widely adopted normalization techniques in deep learning models and we observed significant limitations in both. For batch normalization, we observed high variance for graph embeddings of the same random chosen graph throughout training, which caused stability issues during optimization. For layer norm, it struggled to learn in most graph-level tasks which we hypothesize is related to the oversmoothing effect described in Zhao & Akoglu (2020) and Cai & Wang (2020). We proposed hybrid normalization to address both issues. We found that (1) unlike layer normalization, hybrid normalization maintains high separation between node embeddings throughout training, while (2) variance of node embedding for a given graph across batches is much more controlled compared to batch normalization.

We monitored two key metrics during foundation model training for each of the 3 choices of normalization techniques.

(1) Cosine similarity between neighboring training steps of the node embeddings for the same graph. Let $H_t \in \mathbb{R}^{n \times d}$ be the embedding matrix for a chosen graph at training step $t$, $H_{t,i} \in \mathbb{R}^d$ be the node embedding for node $i$, the metric is defined as

$$\text{cosine\_similarity}(H_t, H_{t-1}) = \frac{\sum_i H_{t,i} H_{t-1,i}}{\sqrt{\sum_i H_{t,i}^2} \sqrt{\sum_i H_{t-1,i}^2}} \quad (1)$$

This measures how much the graph embedding of a given graph changes during training.

(2) Pairwise distances between node embeddings from the output of GNN layers. ("row-diff" measure from (Zhao & Akoglu, 2020) Let $H \in \mathbb{R}^{n \times d}$ be the embedding matrix output by GNN layers, $h_i \in \mathbb{R}^d$ be the $i$-th row of $H$, then row-diff is defined as the average of all pairwise distances between node embeddings:

$$\text{pairwise\_dist}(H) = \frac{1}{n^2} \sum_{i,j \in [n]} \|h_i - h_j\|^2. \tag{2}$$

This is an indicator of how well the GNN can separate node embeddings from each other.

As shown in Figures 3a and 3b, batch norm showed high separability of nodes, however, the cosine similarity of graph embeddings changes drastically during training, which is indicative of unstable learning.In contrast, layer normalization maintained a consistent rate of change for the cosine similarity measure after training stabilized but exhibited lower and narrower node separability, which could be linked to the over-smoothing phenomenon. Hybrid normalization maintained relatively high node separability across training while the variance of cosine similarity appears to be much more controlled than batch normalization.

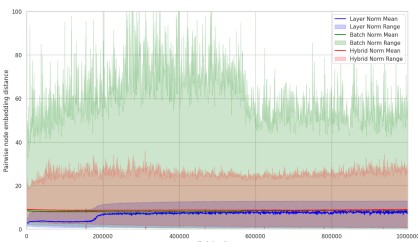

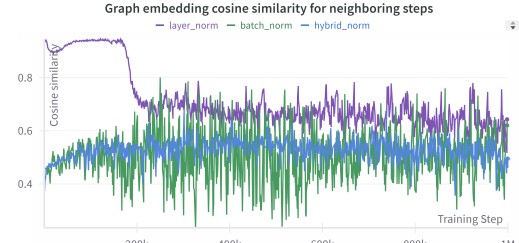

(a) Pairwise distances in node embeddings over batches throughout training.

(b) Cosine similarity between graph embeddings from neighboring training steps.

Figure 3: Pairwise node embedding distance and Cosine similarity for different norms.

# 6 DISCUSSION, LIMITATIONS AND FUTURE DIRECTIONS

Our work demonstrates the promise of foundation models for Boolean logic but is currently limited to small-scale problems, far smaller than typical industrial cases. Working with large instances becomes difficult in multiple ways, it is computationally hard to acquire large enough datasets and the large size of the instances constrains the amount of memory available for an expressive enough network. However, our fine-tuning results suggest that there might be a way to circumvent both of these obstacles. Our results on generalization to unseen tasks with more efficient training suggest that generalizing to unseen problem sizes might also not require massive new datasets. Furthermore, our results using frozen shared representations suggest that fine-tuning only a smaller head network might be sufficient for achieving good performance alleviating the need for massive memory intensive networks at training time.

By solving the hurdles of generalizing to larger problem sizes, we would also expand the diversity of instances we would be able to study. Incorporating a richer class of problem instances could also provide a yet richer shared representation, which could benefit performance even in the regime we currently study. We restricted ourselves to small instance distributions so we could evaluate very large GNN architectures; distributions of small instances, which are difficult across the variety of tasks we study are limited.

Further investigation into the hybrid normalization architecture is potentially exciting. We have empirically demonstrated that hybrid normalization outperforms other normalization techniques when used in isolation, and we have provided a hypothesis to explain these results. It would be interesting to evaluate this method on other graph datasets and to explore the theoretical foundations behind its success.

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

## A  TASK DETAILS

We hereby give the detailed definitions of each task use in model training and evaluation. First, we define a SAT formula $S$ as a set of clauses $C = \{c_1, \ldots, c_m\}$ over a set of variables $V = \{v_1, \ldots, v_n\}$. Each clause consists of a set of Boolean *literals*, defined as either a variable $v_i$ or its negation $\neg v_i$. The set of literals in a clause are joined by OR operators and the set of clauses are joined by AND operators.

### A.1  MODEL COUNTING

For a given SAT formula $S$, its **model count** is defined as the number of distinct truth assignments to variables for which the formula evaluates to true. Formally:

$$\text{ModelCount}(S) = |\{\mathbf{t} \in \{True, False\}^n \mid S(\mathbf{t}) = \text{True}\}|$$

| **Graph-Level Regression Tasks** | | |
| --- | --- | --- |
| **Task** | **Description** | **Output Shape** |
| Model Counting | Predicting the number of satisfying assignments for the SAT instance. | $\phi_{\text{MC}}(S) \in \mathbb{R}$ |
| Instance-wise Structural Features | | |
| – Graph Features | Basic graph statistics (e.g., average degree, clustering coefficient). | $\phi_{\text{GF}}(S) \in \mathbb{R}^{48}$ |
| – Linear Programming Relaxation | Features from LP relaxation of the SAT problem. | $\phi_{\text{LP}}(S) \in \mathbb{R}^6$ |
| – Local Search Probing | Summary stats of probes of saps and gsat solvers. | $\phi_{\text{LS}}(S) \in \mathbb{R}^{22}$ |
| – DPLL Probing | Propagation/depth of dpll probes. | $\phi_{\text{DP}}(S) \in \mathbb{R}^5$ |
| – Survey Propagation Probing | Summary stats from probes of a survey propagation algorithm. | $\phi_{\text{SP}}(S) \in \mathbb{R}^{18}$ |
| **Graph-Level Classification Tasks** | | |
| Predicting Satisfiability | Determining whether the SAT instance is satisfiable or unsatisfiable. | $\phi_{\text{SAT}}(S) \in \mathbb{R}$ |
| **Node-Level Regression Tasks** | | |
| Variable Features | | |
| – Graph Features | Node-specific statistics (e.g., node degree, betweenness centrality). | $\phi_{\text{GFV}}(S) \in \mathbb{R}^{n \times 13}$ |
| – Linear Programming Relaxation | Variable assignments for optimal solution. | $\phi_{\text{NSF}}(S) \in \mathbb{R}^n$ |
| – Local Search Probing | Stats of variable flip counts/weights in local search probes. | $\phi_{\text{LSV}}(S) \in \mathbb{R}^{n \times 10}$ |
| Clause Features | | |
| – Graph Features | Clause node-specific statistics (e.g., node degree, betweenness centrality). | $\phi_{\text{GFC}}(S) \in \mathbb{R}^{n \times 10}$ |
| – Linear Programming Relaxation | Constraint slacks | $\phi_{\text{NSF}}(S) \in \mathbb{R}^n$ |
| – Local Search Probing | clause penalties and frequency satisfied | $\phi_{\text{LSC}}(S) \in \mathbb{R}^{n \times 4}$ |
| **Node-Level Classification Tasks** | | |
| Backbone Prediction | Predicting the backbone status (non-backbone, true, false) of each variable. | $\phi_{\text{BB}}(S) \in \mathbb{R}^{n \times 3}$ |
| Unsatisfiable Core Detection | Predicting probability of variable belonging to unsat core | $\phi_{\text{UC}}(S) \in \mathbb{R}^n$ |
| RL Branching | Predicting MCTS-related properties for each node (visit counts, value estimates). | $\phi_{\text{MCTS}}(S) \in \mathbb{R}^{n \times 2}$ |

Table 5: Categorization of task into graph and node-level regression and classification along with brief descriptions and output shape of each corresponding task head.

## A.2 INSTANCE-WISE GRAPH FEATURES

From Hutter et al. (2014). A SAT problem can be represented as different graph representations. First, a variable-clause graph is a bipartite graph where the two disjoint sets of nodes correspond to variables and clauses. An edge connects a variable node to a clause node if the variable appears (positively or negatively) in the clause. Second, a variable graph have each node corresponds to a variable, and an edge exists between two nodes if the corresponding variables appear in the same

| Graph representation | Metrics |
|---|---|
| variable-clause graph | variable node degree, clause node degree |
| variable graph | node degree, diameter |
| clause graph | node degree, clustering coefficient |

Table 6: Graph metrics

clause. Third, a clause graph have each node corresponds to a clause, and an edge is drawn between two nodes if the corresponding clauses share at least one variable.

We collect different graph metrics in Table 6 from each graph representation, and for each metric compute its mean, variation coefficient, min, max and entropy.

## A.3 INSTANCE-WISE LINEAR PROGRAMMING RELAXATION

SAT instance can also be represented as linear programming problem.

**Variables:**

$$x_j \in [0,1], \quad \forall j \in \{1, \dots, n\}$$
$$s_i \geq 0, \quad \forall i \in \{1, \dots, m\}$$

where $x_j$ is the boolean variable, $s_i$ is the slack variable for clause $C_i$, measuring the degree of under-satisfaction of the clause.

**Objective:** As a constraint satisfaction problem, there's no objective.

**Constraints:** For each clause $C_i$, the following inequality must hold:

$$\sum_{j \in \text{Pos}(i)} x_j + \sum_{j \in \text{Neg}(i)} (1 - x_j) + s_i \geq 1, \quad s_i \geq 0, \quad \forall i \in \{1, \dots, m\}$$

where $\text{Pos}(i)$ is the set of variables that appear positively in clause $C_i$, $\text{Neg}(i)$ is the set of variables that appear negatively in clause $C_i$.

We compute the mean, variation coefficient, min, and max for the Integer slack vector of the LP problem, along with ratio of integer vars in LP solution and objective value of LP solution.

## A.4 INSTANCE-WISE LOCAL SEARCH PROBING

SAPS Hutter et al. (2002) is a dynamic local search algorithm for SAT solving. We run 2 seconds of the solver on the sat instance and record:

- Number of steps to the best local minimum in a run

- Average improvement per step to best local minimum in a run

- Fraction of overall improvement due to first local minimum

- Best solution

## A.5 INSTANCE-WISE DPLL PROBING

From Hutter et al. (2014). DPLL is a fundamental algorithm for SAT solving. A sequence of variable assignments are made until a contradiction is encountered. After each assignment, unit propagation is called, which means literals in a clause by themselves are assigned to true. We make random probes of depth 1, 4, 16, 64 and 256 and measure the number of unit propagations. We also take a number of random probes until a contradiction is encountered and measure average depth.

## A.6 SURVEY PROPAGATION PROBING

From Hutter et al. (2014). Run a survey propagation algorithm Braunstein et al. (2005). Then for each variable, compute the higher of P(true)/P(false) or P(false)/P(true). Then compute statistics across variables: mean, variation coefficient, min, max, 10%, 25%, 50%, 75%, and 90% quantiles.

## A.7 PREDICTING SATISFIABILITY

A Boolean formula $S$ is **satisfiable** if there exists an assignment $\mathbf{V} \in \{\text{True}, \text{False}\}^n$ to the variables $V = \{v_1, \dots, v_n\}$, such that the formula $S$ evaluates to True. Formally:

$$\text{Satisfiable}(S) \iff \exists \mathbf{t} \in \{\text{True}, \text{False}\}^n \text{ s.t. } S(\mathbf{t}) = \text{True}.$$

## A.8 BACKBONE PREDICTION

The **backbone** of a satisfiable formula $S$ is the set of variables that are true in all satisfying assignments of $S$. Formally:

$$\text{Backbone}(S) = \{v \mid \text{variable } v = \text{True } \forall \mathbf{t} \in \{\text{True}, \text{False}\}^n \text{ s.t. } S(\mathbf{t}) = \text{True}\}.$$

## A.9 MINIMAL UNSATISFIABLE CORE

Given an unsatisfiable SAT formula represented in CNF form, a minimal **UNSAT core** of $S$ is the smallest subset of its clauses $C_{core} \subseteq C$ such that $S_{core}$ is unsatisfiable. Formally:

$$\arg\min_{C_{core}} |C_{core}| \ s.t. \ C_{core} \subseteq C, \quad S_{core} = \bigwedge_{c \in C_{core}} c \text{ s.t. Satisfiable}(S_{core}) = False$$

## A.10 RL BRANCHING

MCFS Cameron et al. (2024) is a Monte Carlo Tree Search based algorithm aims to find the optimal branching policy (i.e choosing which variable to search next) for SAT solving. After an offline search and rollouts for SAT instance $S$, we get two measurements from the search tree: variable counts and tree-size estimates. **variable counts** $\in \mathbb{R}^n$ measures how many times each variable was chosen during the search. **tree-size** $\in \mathbb{R}^n$ measures the estimated size of the search tree for each variable.

# B LOSS

For each task $T_i$, the loss $\mathcal{L}_i$ is defined based on the task type $\tau(T_i) \in \{\text{node}, \text{graph}\}$ and category $\ell(T_i) \in \{\text{regression}, \text{classification}\}$. $\tau(T_i)$ defines whether the model makes predictions for every node or a global prediction of entire graph and category $\ell(T_i) \in \{\text{regression}, \text{classification}\}$ defines whether we use mean-squared error or cross entropy loss

$$\mathcal{L}_i(T_i, \phi) = \begin{cases} \mathbb{E}_{S \sim \mathbb{P}} \left[ (T_i(S) - \phi_i(S))^2 \right] & \text{if } \tau(T_i) = \text{graph}, \ell(T_i) = \text{regression}, \\ \mathbb{E}_{S \sim \mathbb{P}} \left[ -\sum_{c=1}^{C} T_i(S)_c \log \phi_i(S)_c \right] & \text{if } \tau(T_i) = \text{graph}, \ell(T_i) = \text{classif}, \\ \mathbb{E}_{S \sim \mathbb{P}} \left[ \frac{1}{N} \sum_{j=1}^{N} (T_i(v_j) - \phi_i(v_j))^2 \right] & \text{if } \tau(T_i) = \text{node}, \ell(T_i) = \text{regression}, \\ \mathbb{E}_{S \sim \mathbb{P}} \left[ \frac{1}{N} \sum_{j=1}^{N} \left( -\sum_{c=1}^{C} T_i(v_j)_c \log \phi_i(v_j)_c \right) \right] & \text{if } \tau(T_i) = \text{node}, \ell(T_i) = \text{classif}. \end{cases}$$

$T_i(S)$ is the ground truth result for task $T_i$ on instance $S$, $T_i(x)_c$ is an indicator if class $c$ is the true class, $\phi_i$ is the task head for task $i$, $\phi_i(x')_c$ is the predicted probability of class $c$ where $x'$ could be a logical formula $S$ or a variable $v_j \in S$. We would like to learn some $\phi$ that takes $S$ or a lossless representation of $S$ directly as input.

## C   SAT ENCODING

A CNF SAT instance $S$ is defined by a set of clauses $C = \{c_1, \ldots, c_m\}$ over a set of variables $V = \{v_1, \ldots, v_n\}$. Each clause consists of a set of Boolean *literals*, defined as either a variable $v_i$ or its negation $\neg v_i$. The set of literals in a clause are joined by OR operators and the set of clauses are joined by AND operators. We represented a CNF SAT instance with $n$ clauses and $m$ variables as an $n \times m$ bipartite graph, where each node and edge is represented as a $d$-dimensional trainable embedding. Variable and clause nodes are represented with embedding vectors $v^0$ and $c^0$ respectively and each edge $(i, j)$ is represented with embedding $e_t$ if the true literal for variable $i$ appears in clause $j$ and $e_f$ if the false literal for variable $i$ appears in clause $j$.

## D   MODEL ARCHITECTURE

**Pre-GNN encoder**  We first map each node and edge embedding with a MLP:
$\forall x : x^1 = \text{MLP}_{\text{pre-node}}(x^0), \forall u, v : e^1_{u,v} = \text{MLP}_{\text{pre-edge}}(e^0_{u,v}), g^1 = \text{MLP}_{\text{pre-global}}(g^0).$

**Message-passing layers**  We take in $(\mathbf{X}^\ell, \mathbf{E}^\ell, g^\ell)$ and output $(\mathbf{X}^{\ell+1}, \mathbf{E}^{\ell+1}, g^{\ell+1})$ as follows:

$$\forall u, v : \bar{e}^\ell_{u,v} = \text{MLP}_{\text{edge}}([x^\ell_u | x^\ell_v | e^\ell_{u,v} | g^\ell])$$

$$\forall i : \bar{x}^\ell_i = \text{MLP}_{\text{node}}\left(\left[x^\ell_i \middle| \sum_{u,i}[\bar{e}^\ell_{u,i}|x^\ell_u] \middle| \sum_{i,v}[\bar{e}^\ell_{i,v}|x^\ell_v] \middle| g^\ell\right]\right)$$

$$g^{\ell+1} = \text{MLP}_{\text{global}}\left(g^\ell \middle| \sum_j \bar{x}^\ell_j \middle| \sum_{u,v} \bar{e}^\ell_{u,v}\right)$$

$$\bar{\mathbf{X}}^\ell = \text{SelfAttention}(\bar{\mathbf{X}}^\ell)$$

$$\forall u, v : e^{\ell+1}_{u,v} = e^\ell_{u,v} + \bar{e}^\ell_{u,v}$$

$$\forall i : x^{\ell+1}_i = x^\ell_i + \bar{x}^\ell_i,$$

where $|$ denotes the concatenation of vectors.

**Shared-embedding layer**  We take the node embedding output of k Message-passing layers $X^k$, pass through a row-wise sum pooling layer to get the graph level embedding

$$emb_{graph} = \sum_{i=1}^m X^k_{i,:}$$

scatter across rows

$$S = \mathbf{1}_m s$$

and concat back to the node embedding $X^k$

$$emb_{\text{shared}} = X^k \,|\, S$$

where $1_m \in \mathbb{R}^m$ denotes a column vector of 1s and $|$ denotes the concatenation of vectors.

**Head networks**  We have a head network for each task in the foundation model and a single head for finetuning and single task training. We first describe the head networks according to task type

- Graph-level tasks: The final prediction is a global sum pooling of all node representations followed by an MLP that maps to output shape according to the specific task: $\bar{y} = \text{Dropout}_p(\text{MLP}_{\text{head}}(\sum_i x_i))$.
- Node-level tasks: The final prediction is an MLP that maps each node embedding to output shape according to the specific task: $\bar{y}_i = \text{Dropout}_p(\text{MLP}_{\text{head}}(x_i))$, where $\text{Dropout}_p$ masks each parameter with probability $p$.

Table 5 in the appendix describes the output space for each of the sixteen task heads.

# E    EXPANDED RESULTS

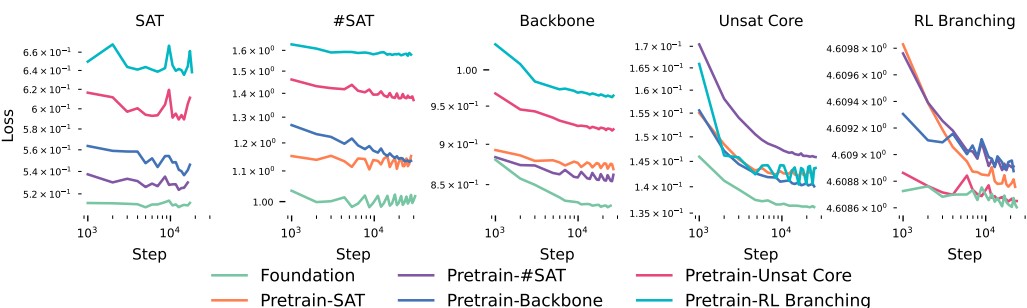

Figure 4: Validation performance over a training run for finetuning from different pretrained models with parameters from the shared architecture frozen.

