# OpenReview forum: "Foundation Models for Boolean Logic"
_ICLR.cc/2025/Conference — Submitted to ICLR 2025_

### Official Review · Reviewer_37GF · 2024-11-01

**Soundness:** 2
**Presentation:** 3
**Contribution:** 3
**Rating:** 6
**Confidence:** 4

**Summary:**

The authors perform multi-task learning to pre-train a GNN for encoding CNF's. The tasks are, for instance, satisfiability, model counting, backbone and unsatisfiable core prediction, and graph statistics on the CNF. The models is trained for 24 hours on a million 3SAT instances with 100 variables. The authors show there is some transfer of the embeddings to held-out tasks.

**Strengths:**

The goal of training a 'foundation model' (or rather: encoder) of CNF formulas for different downstream tasks is novel and could be useful. I personally am quite interested to see where this could go. The paper has a strong, informative introduction, and is overall quite well written.

**Weaknesses:**

The evaluation is quite lacklustre. Here are my three major important issues:
1) There is no empirical comparison to classical solvers. What are the benefits of the model compared to these? Runtime? The paper only compares to other ML methods, but does not make a strong argument for why we should expect 'foundation models' to be more useful / outperform classical solvers.
2) The authors do not perform any out-of-distribution analysis, such as different problem sizes or distributions.
3) Results are poorly presented: Arbitrary choices of precision, loss instead of accuracy where relevant, and no standard deviations.

**Questions:**

- A more detailed analysis of the scaling to larger problem instances would be useful. Since the model involves self-attention, I assume the runtime is quadratic.
- The above also goes experimentally. Furthermore, how does the model perform on datasets from a different generator? (Eg: not at the phase transition)
- The paper does not mention if the code and trained models will be shared, which would be a must for training foundation models
- No scaling experiments are performed. How important is the size of the model for performance?
- The authors report test-loss, which is hard to interpret. For classification, test accuracy is much easier to interpret. For instance, what is the accuracy on SAT? And for the model counting results, the authors should report the average/median true model count in the test set to get an idea of the size of the error. I assume this is low since the instances are 'at the solubility phase transition'.
- How can the model be adapted to perform tasks such as MaxSAT or WMC, which involve weights? And for model sampling (either with weight or uniform)?
- What is the average number of clauses in the dataset?
- I did not understand the pre-GNN-encoder. I think there are 5 things to embed (true/false edge literal, clause, variable, and global), which can be implemented with 5 fixed embeddings. But it's mentioned there are 2 fully-connected layers here.
- What is the Dropout_p for at the output layer? Why would you want to dropout the output? What is the value of p?
- The significance in the tables is arbitrary, randomly going from 2 to 6 digits. More than 2 digits of significance is unnecessary here, especially since no repetitions are performed. Many bolded numbers are not significant in 2 digits, and I would not expect them to hold under repeated runs. Bolding a 0.03% relative improvement is not a good way to communicate your models performance.

---

### Official Review · Reviewer_Uiqv · 2024-11-01

**Soundness:** 3
**Presentation:** 3
**Contribution:** 3
**Rating:** 8
**Confidence:** 4

**Summary:**

This work presents a foundation model architecture for boolean logic tasks. The foundation model is built upon graph neural networks, specifically instantiated from the existing graph transformer GPS++; it is trained on a massive dataset of 3SAT instances and is trained on a wide range of boolean logic tasks, where each type and structure of the tasks are formulated separately in the loss function. Overall, this paper illustrates the capability of the proposed foundation model; the results show that
(1) the foundation model trained with specific held-out tasks can generalize well to those tasks with fine-tuning applied solely on the the output portion of the model;
(2) the fine-tuned foundation models outperforms the single task models;
(3) a hybrid method of applying batch normalization (on the message-passing layers in the graph) and layer normalization (on the head networks) plays a critical role in the success of training the proposed foundation model.

**Strengths:**

- This paper is the first to build a foundation model for a wide range of boolean logic tasks.
- This paper presents a formulation of loss function that incorporates different types and structures of tasks (for example, classification and regression), and shows the possibility to respect all these objectives in the training of a single model.
- Most claims about the model performance are strongly supported by experimental results.
- The related work section gives a clear overview of related prior works.
- The model architecture as well as results are informative and well-organized.
- This work can potentially open up many future directions for building foundation models that target logical reasoning, in which many boolean logic tasks evaluated in this paper play a significant role.

**Weaknesses:**

### In Results
1. Only loss is reported for the performance of the foundation model trained. This does not give a clear intuition on how well might this model behave when it is applied to downstream tasks. It is understandable that regression tasks are usually evaluated on the loss, however, for the classification tasks such as predicting SAT or UNSAT, or model counting where there exists ground truth values, is it possible to include some other performance metric so that it is clear to the reader the power of this trained foundation model? Alternatively, it might also be helpful if the range of the loss can be stated for each of the tasks, so that it is easier to see how well the model is performing in terms of loss.
2. When comparing the fine-tuned models with the models trained from scratch, there seems to be a very minimal difference between the two, especially in the case of these tasks: Backbone, Unsat Core and RL Branching. Claiming that "in all cases the foundation model outperforms the model trained from scratch" might be over-exaggerating.
3. "Instance-wise Structural Features" and "Clause/Variable Features" are introduced as tasks in Table 1. However, there is no results found for these tasks.

### Typos and grammar
There are several typos and grammar errors that should be carefully fixed. Examples include but are not limited to:
1. Line 017 "performed better" should not appear twice,
2. Line 205 missing space after $e_f$,
3. Line 257 has a repetition of an entire sentence,
4. Line 328 should be "otherwise it is satisfiable" in stead of "other it is satisfiable",
5. Line 375 missing space,
6. Line 377 incorrect sentence structure. *Please proof-read carefully*.

### Writing/Notation style and consistency
1. Please be consistent with using "formulas" or "formulae", choose one,
2. In the text, the input to the graph is (X, E, g), but in Figure one, it is shown as (X, E, G),
3. The arrows in Figure 1 should be refined,
4. In the "message-passing layers" section, some notations are confusing. Please see details in the Question section below.

**Questions:**

1. It is quite confusing the way the number of tasks is counted in this paper: in the abstract, it is said to be "eighteen tasks", however, in the introduction, it is said to be "ten tasks".
2. In the methods section, there are a few confusing phrases.
- Line 178 "$\tau(T_i)$ defines whether size of the output is the number if (should be "of") nodes N or a single scalar", what does "a single scalar" refers to? It is pretty clear from the formulas that the authors either apply an average for the node tasks or apply the loss function as is for the graph tasks, but the textual explanation is rather confusing. Is it possible to clarify what is the intended understanding here?
- Line 214 in point (3) there are two "the" (typo), and this sentence also needs a bit more clarifications on its meaning.
- As mentioned in weaknesses, under the section of "message-passing layers" the math formulas presented are a little confusing: where is the computation for $X^{l+1}$? Is that a typo in Line 247?
3. In section 4.1, for the descriptions of the boolean logic tasks, it is not clear whether these computations are executed to obtain the ground truth values? Or are they connected to the foundation model at some point during evaluation? Please clarify.
4. In Table 4, the Rl Branching task does not seem to show a significant difference between the three normalization techniques, is there any hypothesized reason for this finding?

---

### Official Review · Reviewer_syb9 · 2024-11-02

**Soundness:** 2
**Presentation:** 1
**Contribution:** 1
**Rating:** 3
**Confidence:** 5

**Summary:**

This paper proposes a foundation model for Boolean logic, leveraging a series of pre-training tasks to train the model. The training dataset comprises one million random 3-SAT instances. The authors conduct several experiments to investigate the generalization capabilities of the foundational model.

**Strengths:**

The authors claim that this is the first foundation model for Boolean logic and build a large-scale dataset with diverse labeled tasks for SAT problem.

**Weaknesses:**

1. The scale and diversity of the training dataset is very limited. The authors only provide the training dataset consisting of random 3-SAT problems, with each instance only has 100 variables. A foundational model cannot be adequately trained on a single distribution and such small data points.
2. The experiment is not convincing enough, which only focusing on the random 3-SAT problems.
3. The organization of this paper lacks clarity. For example, Line 349 mentions " five foundation models and five single-task models ...", I did not find the formal definition of these five tasks here, the only explanation is provided in Line 81.

**Questions:**

1. A formal definition of these 5 tasks in Table 2 should be provided. In Table 2, the difference in loss values between model counting and RL branching tasks is striking, with model counting losses exceeding 1.0 while RL branching losses hover around 0.04.
2. For Weakness 2, how does the generalization ability of this model to handle other novel distributed problems, like k-colorability or k-clique problems? If not, the proposed model may be overstated, it may be limited to address random 3-SAT problems.
3. What is the motivation of the selection of tasks in Table 1? It appears that the authors collect tasks as much as possible without a clear purpose.
4. It is a commendable attempt to train a foundation model of Boolean logic, however, I do not think current work is solid enough.

---

### Official Review · Reviewer_Fti3 · 2024-11-03

**Soundness:** 3
**Presentation:** 2
**Contribution:** 2
**Rating:** 5
**Confidence:** 4

**Summary:**

This paper focuses on training a foundation model for Boolean logic. The authors employ a Graph Neural Network (GNN) to perform various tasks on CNF-SAT formulas. The core idea is to compare models trained on a single task from scratch with those pretrained on multiple tasks and then fine-tuned for a specific task. The authors also emphasize the significant impact of their chosen normalization technique, HybridNorm, on training stability and convergence. To evaluate their approach, they conduct experiments on randomly generated 3-SAT formulas.

**Strengths:**

The HybridNorm technique can be potentially useful more generally. The results from transfer from other tasks could also be useful but in the current form are not very convincing.

**Weaknesses:**

The paper does not introduce any significant novelties. While the HybridNorm idea is promising, it's not particularly complex. Given that the main contribution lies in demonstrating how new tasks can be learned through fine-tuning a foundation model, the paper could benefit from more extensive experimental results. For example, evaluating the model on different datasets and providing metrics beyond test loss (such as task-specific accuracy) would offer a more comprehensive assessment of the model's performance. I have no sense for how significant the improvement is given that the tables only report test loss.

**Questions:**

1. Why you did not report results in terms of metrics such as accuracy or something which would be more interpretable?
2. At the end of section 1, you mention that you envision models with billions of parameters. Do you think it really makes sense for boolean logic? This makes a lot of sense for language models which need to store large number of facts but it is not clear to me what these billions of parameters would do for boolean logic.
3. I believe the notation in section 3 could be introduced more sensibly. The way you use $\phi$ and $\phi_i$ may be confusing for someone. I understand that $\phi$ corresponds to a body of the model $\phi_i$ corresponds to a body with a head but I think it would be better to mention it explicitly.
4. It would be nice to know more details about the creation of the dataset, i.e. how long on average it takes for each solver to solve individual tasks. Better description of the individual tasks would also be useful because in the current form it's not reproducible. This would be a good section to the appendix.
5. I do not understand the last sentence in section 4.1 " These may be alternatively useful as features for classification such as adding to the structure based variable-level features developed by Zhang et al. (2024)." I believe rewriting it more clearly would help the reader.
6. In  5.1 you mention that "there existed a discrepancy between the batch statistics during training and validation." Why is it the case? Shouldn't these sets be from a same distribution? Or do you want to say that in inference time you don't use batching?

---

### Meta-Review · Area_Chair_uha7 · 2024-12-20

**Metareview:**

The paper focuses on end-to-end Boolean logic modeling. To this end, it suggests a Graph Neural Network (GNN) to perform various tasks on CNF-SAT formulas. Overall, the reviews agree that this is an interesting research direction, namely, covering a wide range of Boolean logic tasks. Unfortunately, the reviews also present salient arguments about the suitability of this paper for ICLR in its current form. One reviewer points out a weak novelty, as the HybridNorm idea is promising but "not particularly complex". Another reviewer points out a potential downside that, while the data set consists of 1 million random 3SAT instances, each instance only has 100 variables. This seems to be in contrast to evaluating SAT solvers (see e.g. https://www.ijcai.org/Proceedings/15/Papers/058.pdf). One other reviewer points out that a comparison to classical solvers is missing. To summarize, a nice direction but some downsides have to be addressed before publication. Please note that this should not be taken as a statement regarding the usefulness of your research.

**Additional Comments On Reviewer Discussion:**

he discussion arose from issues raised in the reviews. The feedback was acknowledged by one reviewer, saying that it addressed his concerns. Moreover, the reviewer says that this is a very interesting work, but the experimental evaluation is too weak. The other discussion did not change the opinion of the reviewer. This was about statistical significance, readability, and OOD, among other.

---

### Decision · Program_Chairs · 2025-01-22

Reject